# FOODCAM: A Novel Structured Light-Stereo Imaging System for Food Portion Size Estimation

**DOI:** 10.3390/s22093300

**Published:** 2022-04-26

**Authors:** Viprav B. Raju, Edward Sazonov

**Affiliations:** Department of Electrical & Computer Engineering, The University of Alabama, Tuscaloosa, AL 35487, USA; vbraju@crimson.ua.edu

**Keywords:** food portion, portion size estimation, food volume, food imaging, dietary assessment

## Abstract

Imaging-based methods of food portion size estimation (FPSE) promise higher accuracies compared to traditional methods. Many FPSE methods require dimensional cues (fiducial markers, finger-references, object-references) in the scene of interest and/or manual human input (wireframes, virtual models). This paper proposes a novel passive, standalone, multispectral, motion-activated, structured light-supplemented, stereo camera for food intake monitoring (FOODCAM) and an associated methodology for FPSE that does not need a dimensional reference given a fixed setup. The proposed device integrated a switchable band (visible/infrared) stereo camera with a structured light emitter. The volume estimation methodology focused on the 3-D reconstruction of food items based on the stereo image pairs captured by the device. The FOODCAM device and the methodology were validated using five food models with complex shapes (banana, brownie, chickpeas, French fries, and popcorn). Results showed that the FOODCAM was able to estimate food portion sizes with an average accuracy of 94.4%, which suggests that the FOODCAM can potentially be used as an instrument in diet and eating behavior studies.

## 1. Introduction

Accurate dietary monitoring is necessary in this day and age with a high incidence of obesity and other diet-related-chronic diseases. An estimated 93.3 million adults and 13.7 million children and adolescents in the U.S. are affected by obesity according to the Center for Disease Control and Prevention (CDC) [1]. Obesity increases the risk for health problems such as type 2 diabetes, heart disease, and certain types of cancer. Research shows that prevention efforts early in life reduce the risk of obesity later [2]. A recent study [3] revealed that obesity increases the risk for more severe complications of COVID-19. Thus, the development of accurate and reliable tools for dietary assessment is critical for both the general population and those with special nutritional needs, who would most benefit from professional help.

Tracking and monitoring the total energy intake of an individual is essential. Energy intake is dependent on energy density (kcal/g or kcal/mL) and food portion size (g or mL) [4]. Energy density can be calculated by food type recognition (either conducted by computer vision or manually by the user) followed by a simple database search. Estimating the food portion size, however, is a challenging task. FPSE using traditional methods involve estimation by directly measuring the quantity of food (mass/volume) using household measures or visual approximations, for example, estimating the volume in cups. These methods are typically used in food records, 24-h recall, and/or digital food records [5]. The traditional methods are cumbersome, subject to memory (and therefore prone to error), and inaccurate. A recent review on sensor-based FPSE emphasized the extreme importance of the accuracy of portion size estimation for the accurate measurement of energy and nutrient intake [6]. Sensor-based FPSE may be based on imagery [7,8,9,10,11], hand gestures [12], and/or chews and swallows [13,14].

The majority of the sensor-based FPSE methods are imaging-based. Imaging methods can be separated based on the number of viewpoints: single view or multi-view. The popular single view approaches include geometric-models [15], neural nets [16], VR based referencing [17], circular object referencing [10,18], and thumb based referencing [19]. Most single-view approaches need visual cues in the images as a dimensional point of reference [10,19,20,21]. The type of visual cue used determines the complexity of the setup. Some methods require users to carry around a dimensional reference (checkerboards, blocks, cards), increasing user burden. Stereo-imaging-based 3D reconstruction has been previously used for FPSE [8,22,23,24]. A study [22] explored the stereo reconstruction approach by considering real food items. The authors claimed that the stereo approach works well for FPSE with an average error in volume of 5.75 (±3.75)%. However, the method had issues with textureless objects and varying lighting conditions. Another study [25] used a wearable stereo camera system to estimate volumes of 16 types of food objects with error rates of 3.01% in food dimensions and 4.48% in food volume. A similar study [26] used a wearable stereo-camera and achieved an error rate of 2.4% in volume estimation. Both studies [25,26] worked well for irregular shaped food items. However, two-view stereo wearable devices were not tested in free-living and may not be a viable option.

A popular approach in 3D reconstruction and single-view volume estimation methods is structured light reconstruction [27,28,29]. Structured light approaches work by projecting a distinct light pattern onto a scene to acquire a depth map, typically using a single camera and a single projector [30].

Multi-view approaches obtain the three-dimensional geometric information of an object from multiple images. Stereo-3D reconstruction and volume estimation is one such approach. Stereo images acquired by two cameras or by a single camera at two viewing angles are used to determine the 3-dimensional world coordinates of an object [8,26].

Table 1 summarizes the advantages and disadvantages of these two popular 3-D reconstruction techniques. In [31], the authors combined the two methods for acquiring high-complexity stereo image pairs with pixel-accurate correspondence information using structured light. The features of the structured light are matched instead of pixels. This study proposes combining a stereo camera and an infrared (I.R.) dot projector for FPSE.

The contributions of this paper are: (i) a novel FPSE methodology that combines stereo vision and structured light techniques that have not been used previously in the field of dietary assessment; (ii) implementation of a stand-alone monitoring device that enables passive, multispectral, motion-activated, structured light-stereo vision-based food intake monitoring (FOODCAM) with specified spatial resolution requirements; (iii) the FPSE methodology that is accurate for bulk foods and irregular-shaped food items; (iv) the FOODCAM provides high resolution RGB images for other tasks of dietary assessment unlike in other depth modules; and (v) the FOODCAM does not interfere with the activities of the user and can be placed at a comfortable height above the eating surface.

The rest of this paper is organized as follows. First, the device configuration and device characteristics are discussed in Section 2, along with a description of the methodology. Section 3 summarizes the results, and Section 4 provides an in-depth discussion about the current approach, followed by the conclusions in Section 5.

## 2. Materials and Methods

### 2.1. Device Description

The FOODCAM device is depicted in Figure 1. The device consists of an STM32 processor, two OV5640 cameras (with an electromechanically switched infrared-block filter), each with an FPGA + 1 MB RAM as a frame buffer, an ADXL362 accelerometer, a Panasonic EKMC1601112 PIR sensor, an IR dot projector, and a micro S.D. card to store the images. The filters were fitted with 90 degrees (horizontal field of view) wide-angled lens. The device can store ~1 million images. The device has a 3000 mAh battery and has an average runtime of 14 h (continuous image capture mode). FOODCAM captures four images (two stereo pairs of RGB and RGB + IR) every 10 s.

A mechanical lens switch shifts between a transparent filter and an IR block filter and is synchronized with the IR dot projector. The transparent filter is active when the I.R. pattern is projected, and the cameras capture an IR stereo-image pair. The IR block filter is active when the IR pattern is off, and the cameras capture a RGB stereo-image pair.

The device has two capture modes: continuous and motion-activated. The PIR sensor can be used to trigger image capture only when motion is detected and thus saves the battery life. The FOODCAM device was primarily built to monitor food intake and cooking activities in kitchens or dining areas..

### 2.2. Design for Specified Spatial Resolution

The FOODCAM was designed with consideration of the spatial resolution including depth resolution, or the accuracy with which changes in the depth of a surface can be estimated [32].

The following equation represents depth resolution Δz:(1)$$\Delta z={\;z}^{2}\frac{\Delta d}{f.b}\;$$
wherez is the depth of the object from the stereo system;f is the focal length of the cameras;b is the baseline distance between the two cameras; andΔd is the disparity resolution.

Disparity resolution is the precision of disparity estimation and is directly proportional to pixel size, where smaller pixel sizes provide better resolution. Typically, the disparity can be estimated accurately to about one-fifth of a pixel [28]. The camera parameters needed to calculate depth resolution are given below:Pixel Size for OV5640 = 1.4 μm square;Disparity resolution, Δ*d*: 1/5 ×1.4 μm = 0.28 μm; andEffective Focal Length = 3.38 mm.

Equation (1) was used to calculate the depth resolution as a function of depth for different baseline distances. Depth values in the range of 121.92 cm (4 ft) to 243.84 cm (8 ft.) and baseline distances in the range of 3.04 cm (0.1 ft.) to 24.3 cm (0.8 ft.) were considered. A baseline distance of greater than 152.4 mm (0.5 ft.) was identified as the optimal value to achieve a depth resolution of at least 1.5 mm. A baseline of 152.4 mm was selected.

Once the FOODCAM was designed and fabricated, the optimal height at which the FOODCAM should be installed was determined by considering the area that can be covered by the device and the pixel resolution (Table 2):Pixel resolution: The device was used to capture images of a checkerboard image at different distances from the camera. The checkerboard image had squares of unit size (1 cm sides). From this set of images, the parameters described in Table 2 were determined. The number of pixels in each unit square (pixels/square cm) was determined from the images from which the size of each pixel was calculated (sq. mm/pixel). The size of each 2-dimensional pixel provides the spatial resolution of the camera. This measurement was taken prior to camera calibration. A pixel resolution of around 0.5 mm/pixel was set as a requirement.Area: Table 2 also includes the area covered by the FOODCAM as a function of height. The area of overlap of the two cameras is the area covered by the FOODCAM. The area covered by the FOODCAM was practically measured by capturing images at different distances from the floor. An area of at least 106.68 cm × 106.68 cm (3.5 ft. × 3.5 ft.) was set as a requirement.

The optimal height was selected as 152.4 cm. At this range, we had a depth resolution of 1.26 mm and a horizontal pixel resolution of 0.55 sq. mm per pixel. This height was used as a reference while positioning the FOODCAM. FOODCAM was tested by mounting the device on top of a table with the camera baseline parallel to the ground. The camera was positioned at approximately 5 ft. (152.4 cm) from the surface of the table. However, the height of installation cannot always be guaranteed to be 5 ft. In this study, we assumed that the height would be approximately 5 ft. and the pixel resolution was 0.55mm. This condition would restrict the FOODCAM from being used at different distances.

### 2.3. Calibration and Stereo Rectification

Figure 2 describes the proposed framework for the processing of FOODCAM data. Calibration is the first step of this proposed framework. The stereo camera has to be calibrated to eliminate errors due to misalignment of imaging sensors and/or lens distortion. Geometric calibration of the stereo camera was performed as in [33,34] with a checkerboard. Barrel distortion due to lens curvature was removed as in [35]. The calibration for the FOODCAM was conducted using RGB images captured from the device.

### 2.4. Semi-Global Matching

Stereo matching is the process of finding correspondence between the pixels of the stereo images. The semiglobal matching (SGM) method [36] was used as the stereo matching algorithm. SGM is based on the idea of pixelwise matching of mutual information and approximating a global, 2-D smoothness constraint by combining many 1-D constraints.

The aggregated (smoothed) cost S(p,d) for a pixel p and disparity d was calculated by summing the costs of all 1D minimum cost paths that end in pixel p at disparity d. This cost was optimized for each horizontal search of correspondence between the stereo image pairs (using the RGB + IR image pair containing the structured light pattern). The computed disparity map was then used for volume reconstruction.

### 2.5. Gaussian Interpolation and Median Filtering

There can be discontinuities in the disparity map even after a dense pixel matching is performed. The disparities can arise due to occlusions or due to noisy pixels.

Gaussian interpolation uses the 2-D Gaussian distribution as a point-spread function. This fills up disparity discontinuities by convolution with the 2-D distribution. After the interpolation, median filtering is used to remove grainy, speckle noise and provide a smooth map.

### 2.6. 3D-Reconstruction

The estimated disparity map must now be converted from 2-D points to 3-D world coordinates. Triangulation is the process used to project the points in the 2-D images to world coordinates given the disparity map and the geometry of the stereo camera. The geometry of the stereo camera is obtained in the calibration step. Triangulation can be described as given below [36]:

In Figure 3, the point P is a 3-D point in world coordinates. Point P is observed at points P_l_ and P_r_ in the left and right image planes, respectively. We assumed that the origin of the coordinate system coincided with the center of the left imaging sensor. From triangular similarity (ΔPMCl~ΔPlLCl), we obtain:(2)$$\frac{x}{z}=\frac{x_{l}^{\prime }}{f}$$

Similarly, from the similar triangles ΔPNCr and ΔPrRCr, we obtain:(3)$$\frac{x-b}{z}=\frac{x_{r}^{\prime }}{f}\;$$

From Equations (2) and (3), we obtain:(4)$$z=\frac{\mathrm{bf}}{x_{l}^{\prime }-x_{r}^{\prime }}$$

Thus, the depth at various points in the image may be recovered by learning the disparities of corresponding image points. Once z is obtained, x and y can be calculated, and the 2-D image point can be projected in world coordinates. Since we have a dense disparity map, we can obtain a dense point cloud.

### 2.7. Volume Estimation (for FOODCAM)

After the point cloud is obtained, integration can be used to obtain the volume of the object. A voxel is the smallest 3-D unit that can be measured (similar to a pixel in 2-D). We assumed that each point in the point cloud surface represents one voxel bar consisting of many unit voxels. The surface area of one voxel can be obtained by multiplying the unit lengths in the x- and y-axes. The volume of one voxel bar can be obtained as follows:(5)$$V_{vbar}=A_{voxel}\times h_{vbar}\;$$
where *h_vbar_* is the height of each voxel bar as measured from the 3-D depth map. The surface of the dish or plate in each point cloud is considered as the base or bottom of the object. The surface area of the voxel is the square of the pixel resolution (0.5 mm × 0.5 mm) in the x- and y-axes. The height is the distance from the base of the object to the top of the voxel bar.

Once the volumes of all the voxel bars are calculated, we can obtain the volume of the object as given:(6)$$\mathrm{Volume}\;\mathrm{of}\;\mathrm{Object}=\overset{n}{\underset{i=1}{\sum }}V_{vbar,i}\;$$

### 2.8. PMD CamBoard Pico Flexx

The CamBoard Pico Flexx is a 3D camera development kit based on pmd Time-of-Flight (ToF) technology. It is a slim external USB device, about the size of a pack of gum, for flexible 3D sensing use cases using a VCSEL based IR illumination (Laser Class 1). X, Y, and Z values for every pixel resulted in a point cloud of the observed scene. In addition to the Z value, every pixel provides a gray value, which represents the signal strength (amplitude) of the active illumination, so this is an IR gray value image. It can be used for standard 2D image processing, and is perfectly aligned to the depth image. It is also not affected by background light, so it is a very robust 2D image in every light condition. This data also directly correspond to the depth data quality, so it gives a performance indication for the depth data. This depth camera was placed at approximately five feet to capture data to compare to the FOODCAM. The Pico Flexx camera depth images, once captured, were processed using MATLAB 2020B. The grayscale images were used to segment the region of interest, and then the final point clouds were obtained. The volumes of these point clouds were calculated using the alpha shape [37] watertight model for point clouds. An alpha shape creates a bounding volume that envelops a set of 3-D points.

## 3. Results

Five NASCO plastic food models—French fries, popcorn, chickpeas, chocolate brownie, and banana (Figure 4)—were selected to validate the proposed volume estimation method. The volume of each food item was measured using the water displacement method. Each food item was placed on a food tray.

The test objects (with the tray) were placed at random positions on the table (Figure 5). At each position, four sets of stereo RGB and RGB + IR images were captured from the FOODCAM and four sets of depth and grayscale images were captured using the Pico Flexx cam. Samples of RGB and RGB + IR stereo image pairs are shown in Figure 6. Sample grayscale image captured from Pico Flexx camera is shown in Figure 7. The RGB + IR image pairs were used to obtain the food volume. RGB images can be used to identify the food items and used as masks for 3D models to better identify regions of interest in the image scene. Figure 8 depicts stereo image rectification. It can be seen in the image that the unrectified images were not aligned vertically. This misalignment is caused by the manufacturing of the FOODCAM device. Once the images are rectified, they are processed using the SGM algorithm for disparity estimates. Figure 9 depicts the disparity map obtained for a sample set for French fries.

A trained research assistant manually segmented the region of interest using the RGB images. The tray was selected as the region of interest. Each region of interest was then separated into the background (tray) and foreground (food item) object using the point clouds.

The height (hvbar) of each voxel of the food item was then calculated from the top of the tray. The volume was then calculated as in Equation (6).

Figure 10 presents the segmented point clouds obtained for the five food models. Table 3 summarizes the results of the volume estimation from the Pico Flexx cam. The mean absolute error in volume estimation for the five test items was 16.62%. Table 4 summarizes the results of volume estimation from the FOODCAM. The mean absolute error in volume estimation for the five test items was 5.40%.

## 4. Discussion

FOOODCAM device is based on a novel approach to FPSE. It combines two state-of-the-art technologies in depth/3-D reconstruction.

The advantages of the proposed approach are five-fold:The IR projector provides an artificial texture that facilitates the stereo matching algorithm for food scenes, where major portions of the image may be flat and texture-free (e.g., plate or table surface). In traditional methods, matching accuracy suffers due to ambiguities in matched pixels on such surfaces.The problem of matching the structured light from the projector and the light pattern projected in the image, as in the case of structured light reconstruction, is replaced by a more straightforward stereo-correspondence problem, allowing the use of a random projection pattern, and thus, a less expensive projector.The projector or the pattern in the proposed method does not need calibration. Any random pattern can be used, thus reducing the cost and complexity of the projector being used.The proposed approach does not require any fiducial markers.Once the device is calibrated, it can be fixed to a location to monitor food intake. The same calibration parameters can be stored and re-used for that device. In other words, the calibration only needs to be conducted once.

FOODCAM has a unique characteristic of capturing two separate stereo image pairs. The RGB + IR pair is primarily used for portion size estimation using structured light-stereo vision-based volume estimation. The RGB pair can be used for other dietary assessment purposes such as food type recognition. The FOODCAM design conforms to specific spatial resolution requirements. FOODCAM provides a resolution of 0.55 mm in the (x,y) plane and a resolution of 1.26 mm in the z-direction (voxel resolution) at the optimal height of installation.

The current study was conducted using the continuous image capture mode. FOODCAM, unlike other 3D scanners, can capture a temporal sequence of images and possibly be used to construct a temporal sequence of 3-D scans. The temporal aspect of the FOODCAM can be used to track food intake from the beginning of the meal to the end of the meal. This way, the FOODCAM may be used to detect intake behavior such as eating rate as well as to estimate the actual food amount consumed (by considering the portion of leftovers). The motion-activated mode was primarily included to increase battery life. The FOODCAM has a battery that lasts for more than a day and can be used in remote locations without any additional wiring.

Another possible use of FOODCAM is to monitor cooking. It can be used to track the meal preparation from start to end, identifying ingredients and the recipe. The multispectral aspect of the FOODCAM may also be used to detect energy dense items as in [38] and could be one possible future work.

The FOODCAM is a passive device and can be used in wild, free-living conditions. However, the FOODCAM is intended to be positioned straight above the region of interest, which provides an overhead view of the scene. This position/orientation can be difficult to achieve all the time. One possible solution is to use a tripod or support system onto which the device can be mounted. The device has been tested at indoor locations outside the laboratory. The indoor locations all had closed rooms with indoor artificial lighting with minimal sunlight. How the device will behave in outdoor locations or in locations with plenty of sunlight is not known. Testing under different lighting conditions could be a possible future direction.

An important point to consider is that the FOODCAM provides a dense point cloud. Most of the FPSE methods that were proposed previously suffered while testing irregularly-shaped food items. The dense disparity maps and point clouds obtained from the FOODCAM were robust and work well on irregular shaped or bulk plastic food models. Future work could include real foods. A major concern of using FOODCAM is the noise caused by bright reflective spots in the images. It was seen that bright spots on the table produced noise in the disparity map (Figure 6).

The design section of the paper discusses the ideal design parameters for the functioning of the proposed two-camera approach. The area and the pixel resolution are practically estimated from the camera. We then identified the optimal height of installation for the camera using these measurements. We identified 152.3 cm (from the eating surface) to be optimal for our design considerations. However, we cannot guarantee that the camera will be accurately placed at the desired height at all times. We used this distance as a guideline for our processing and assumed that the pixel resolution was 0.55 mm at all depths, irrespective of how big the food item was.

We noticed that the error in volume estimation was not linear across multiple trials. This is essentially because of the variations or fluctuations in the IR projector and the lighting conditions. Since there was a delay between image capture in the left and right cameras, the image capture was not at the same instant, unlike in other stereo-cameras. This delay induced variations in the light and IR pattern intensity in the RGB + IR images. These variations were translated into variations in the final volume estimation, which was preceded by the stereo-matching algorithm.

Another point to be noted is that the FOODCAM needs to be positioned parallel to the surface, looking straight down to achieve optimal performance. This is not always possible, but is acceptable for indoor locations. Additionally, the optimal distance to the eating surface should be approximately 152.4 cm (5 ft), as calculated. This is a limitation since the FOODCAM needs a specific setup for optimal use. In [39], the authors discussed a similar setup for monitoring food intake. The study utilized the universal eating monitor (UEM) that was previously tested under restricted eating conditions, in order to detect and measure individual bites. The paper described a new algorithm to eliminate the limitation of previous studies. A table-embedded scale to be used to measure individual bites during unrestricted eating proved to be a potential alternative to traditional FPSE. Similarly, the FOODCAM can potentially be used during unrestricted eating, and future work could include these analyses.

The results indicated that the picocam provides high precision (low values for std. dev in error in volume estimation). There are numerous other devices such as the picocam that can be used for FPSE. However, one immediate point to be noted is the power consumption of these products. The FOODCAM can last for at least 12–14 h in the continuous capture mode and up to three days in the motion-activated capture mode. Additionally, the device can store data captured during the entire duration of this period on the device memory, therefore, making the FOODCAM a passive and standalone device. The FOODCAM can also provide high resolution RGB images unlike in other depth modules and therefore, is not only capable of depth estimation, but can be used for other tasks of dietary assessment.

A limitation of this paper is the quantity and variety of trials to test the accuracy of the devices. Only five food items were tested in this study, which does not replicate the variety of foods in reality. The number of food items (five) and number of measurements for each food item (12) may not be enough to validate the use of FOODCAM for daily use in FPSE. Future work may include the use of FOODCAM in varying conditions and the test set could include a larger number of food items and more number of measurements to test the use of this device in free-living and real-world scenarios.

At the present moment, image segmentation is conducted manually by a trained assistant. Image-based food segmentation, followed by image-based food recognition, are important steps in dietary assessment; however, our current focus is on the estimation of portion size. Energy density can be calculated by food type recognition (either conducted by computer vision or manually by the user), followed by a simple database search. Segmenting food items is a major step in the process, and we intend to automate it in the future; however, this remains a major limitation of this paper. In the future, this additional step could be eliminated, leading to fully automated FPSE. A possible solution is to include food segmentation to identify the food item/ object of interest. Additionally, the approach used to estimate volume assumes that the objects have a flat bottom. Food items and other objects that have concave bottoms may introduce errors in estimation.

## 5. Conclusions

In this study, a novel approach for FPSE using imaging sensors was proposed. The method combined two state-of-the-art volume estimation methods and did not require fiducial markers. The FOODCAM was tested using five NASCO plastic food models. The mean absolute error in volume estimation was 5.60%, suggesting that the FOODCAM can be used as a reliable method for FPSE.

## Figures and Tables

**Figure 1 sensors-22-03300-f001:**
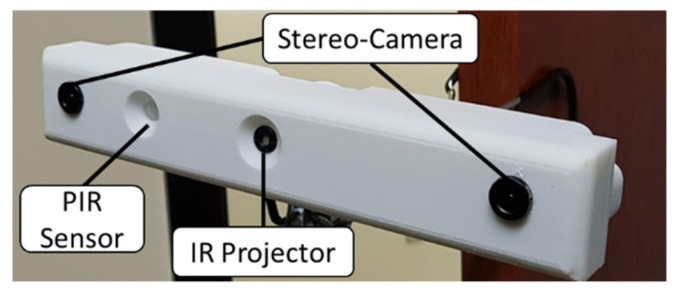
FOODCAM device with a stereo camera, a PIR sensor, and an infrared projector.

**Figure 2 sensors-22-03300-f002:**
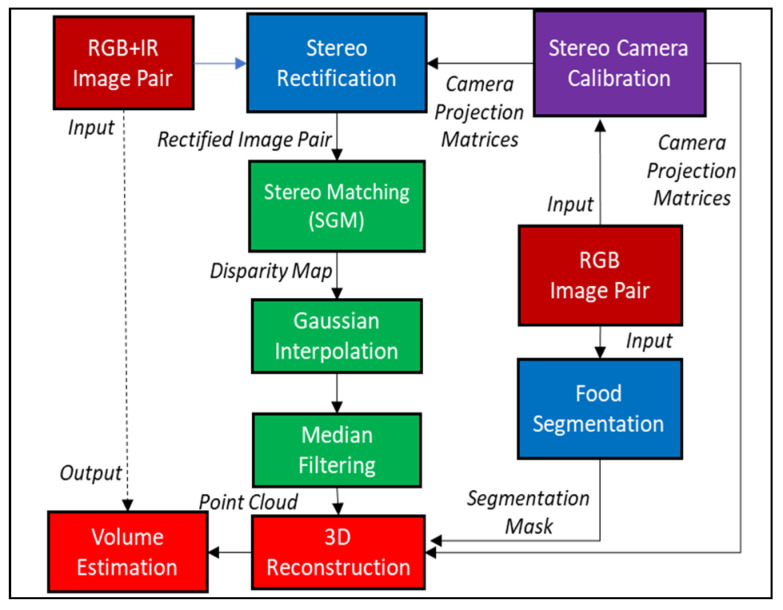
Flowchart of the proposed approach.

**Figure 3 sensors-22-03300-f003:**
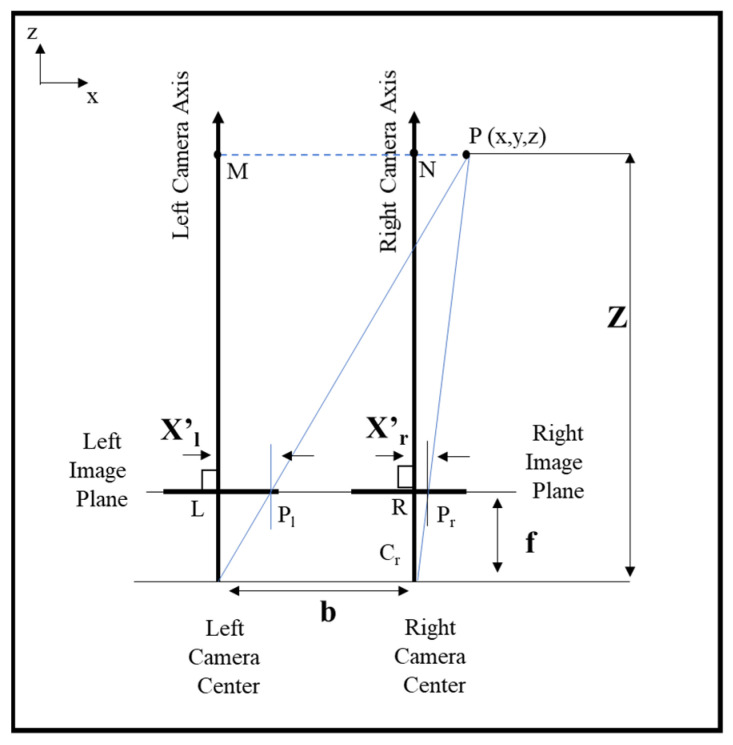
The above plot illustrates how projections of a 3D-point on the x-axis of the stereo-image is used to estimate real-world coordinates. P is a 3-D world point with coordinates (x, y, z). C_l_ and C_r_ represent the left and right cameras. L and R are the principal points (origin) of the left and right cameras, respectively. z_l_ and z_r_ are the z-axes that are normal to the image plane of the two cameras. P_l_ and P_r_ are the projections of the point P on the *x*-axis of the images in the left and right cameras, respectively. X’_l_ is the distance between the principal point L and the projection point P_l_. Similarly, X’_r_ is the distance between the principal point R and the projection point P_r_. M and N are the projections of P on z_l_ and z_r_, respectively. f is the focal length, and b is the baseline distance of the stereo camera. X’_l_ and X’_r_ are used to estimate the disparity between the projections of point P on the left and right image planes. Finally f, b, and disparity are used together to obtain the real-world depth z of the point.

**Figure 4 sensors-22-03300-f004:**
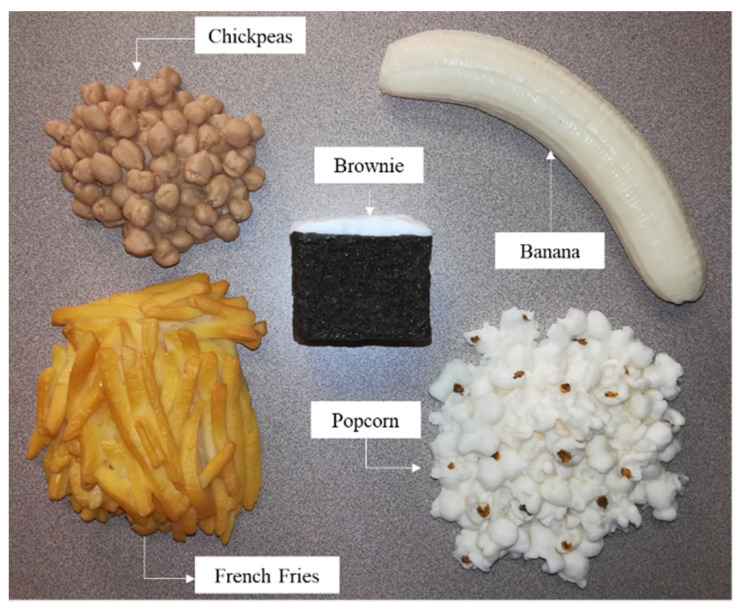
NASCO food replicas.

**Figure 5 sensors-22-03300-f005:**
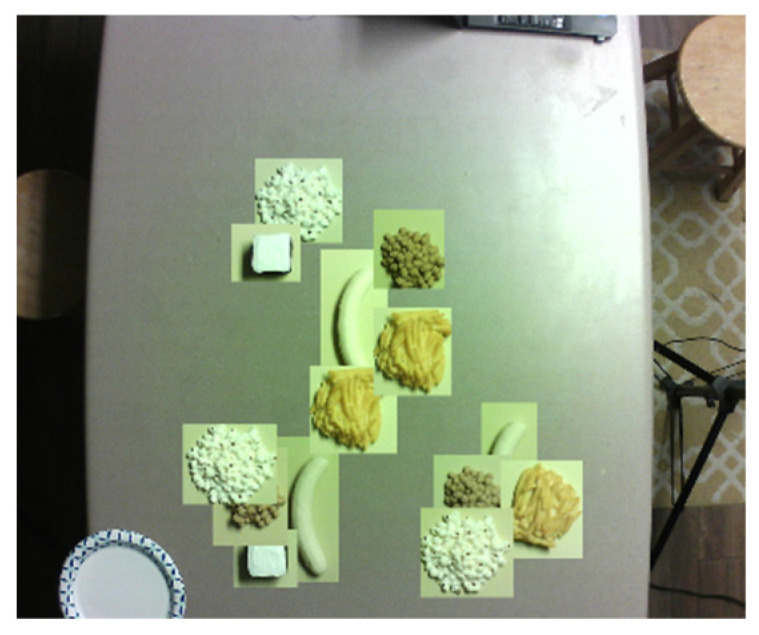
Random positioning of the test items on the table. The area covered by the FOODCAM was predetermined by capturing a test pair.

**Figure 6 sensors-22-03300-f006:**
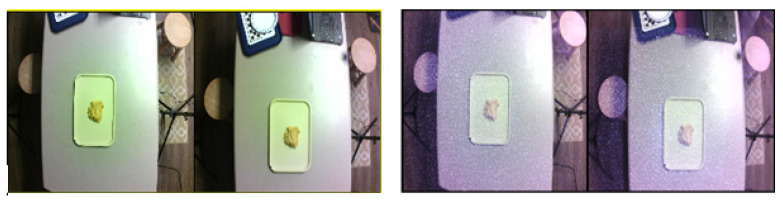
(**Left**) RGB stereo images captured by the FOODCAM (unrectified); (**Right**) RGB + IR stereo images captured by the FOODCAM (unrectified).

**Figure 7 sensors-22-03300-f007:**
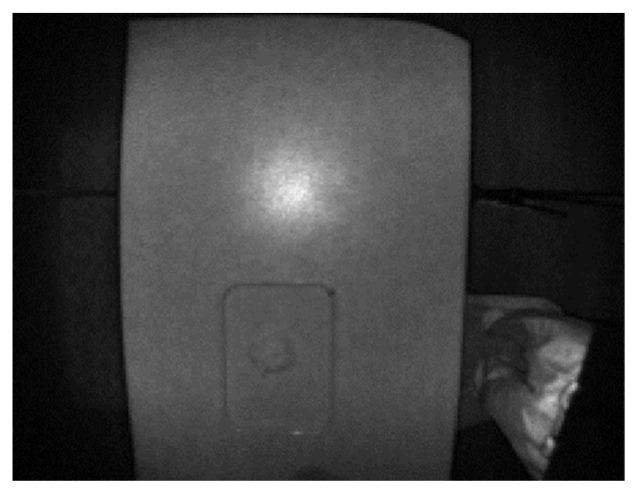
Grayscale image capture by the Pico Flexx camera.

**Figure 8 sensors-22-03300-f008:**
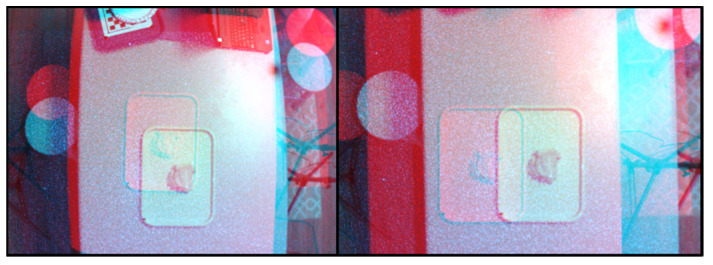
Stereo-image rectification: (**Left**) unrectified images (red-cyan composite view); (**Right**) rectified images (red-cyan composite view).

**Figure 9 sensors-22-03300-f009:**
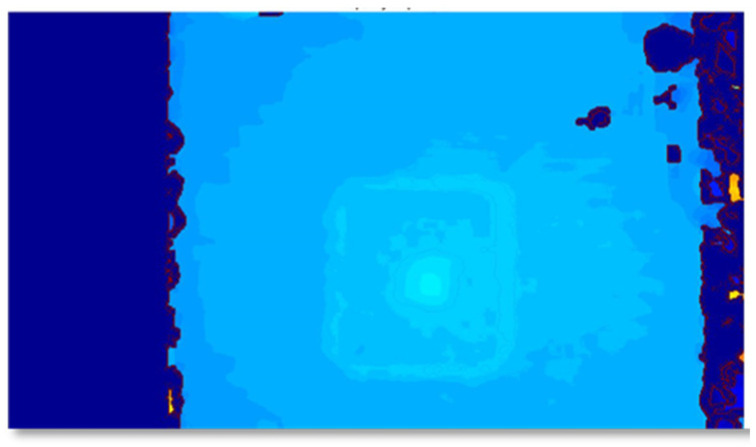
Disparity map obtained after stereo matching the left image with respect to the right image; the color bar represents the disparity level.

**Figure 10 sensors-22-03300-f010:**
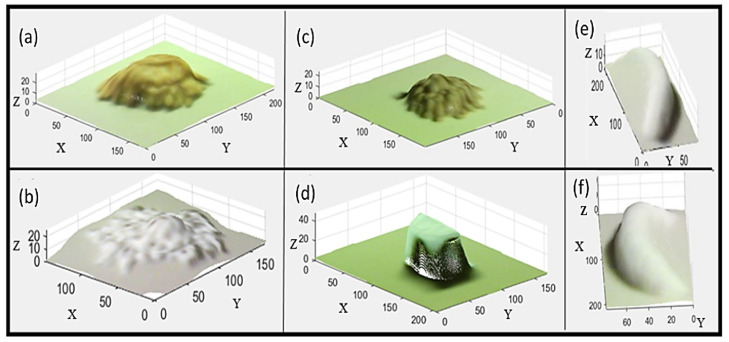
ROI point clouds—oblique view: (**a**) French fries, (**b**) popcorn, (**c**) chickpeas, (**d**) chocolate cake, (**e**) banana (oblique view—1), (**f**) banana (oblique view—2). All units are in mm.

**Table 1 sensors-22-03300-t001:** Comparison between stereo and structured light reconstruction techniques.

Criterion	Stereo Reconstruction	Structured Light Reconstruction
Number of Viewpoints	Two (stereo camera or 1 camera at two view angles)	One (1 camera and a structured light projector)
Advantages	(1) Provides a depth perspective of an object without any information about the surroundings. (2) Eliminates fiducial markers (3) Once cameras are calibrated, the pixel correspondence problem is reduced to a horizontal search	(1) Adds texture to objects. (2) Reduces the number of viewpoints needed. (3) Accurate and dense pixel- correspondences can be automatically produced.
Limitations and Disadvantages	(1) Occlusions (2) Correspondence problem in case of texture-less objects	(1) Projector needs to be calibrated, and the pattern must be known. (2) Computations can be slower and time-consuming

**Table 2 sensors-22-03300-t002:** Pixel resolution of OV5640.

Height of Camera Installation (cm)	mm/Pixel	Pixels/sq. cm	Area Covered
60.96	0.14	4900	69.92 cm × 69.92 cm
91.44	0.28	1296	76.2 cm × 76.2 cm
121.92	0.40	676	101.92 cm × 101.92 cm
152.4	0.55	400	112.78 cm × 112.78 cm
182.88	0.71	196	128.02 cm × 124.97 cm
213.36	0.83	144	158.5 cm × 158.5 cm
243.84	1	100	179.83 cm × 182.88 cm
274.32	1.2	64	207.27 cm × 207.27 cm
304.8	1.4	36	236.74 cm × 236.74 cm

**Table 3 sensors-22-03300-t003:** Errors in volume estimation using the Pico Flexx cam.

Food Item	Random Position	Predicted Volume (mL)				Mean Predicted Volume (Mean ± Std. Dev.)	Ground Truth	Mean Error in Volume Estimation (Mean ± Std. Dev.)
		Trial 1	Trial 2	Trial 3	Trial 4			
Chickpeas	1	77.58	75.2526	75.2526	78.3558	75.84 ± 2.07 mL	100 mL	−24.17 ± 2.01%
	2	76.0284	76.8042	74.4768	80.6832			
	3	73.701	74.4768	73.701	73.701			
French Fries	1	147.54	144.5892	146.0646	141.6384	146.66 ± 3.42 mL	180 mL	−18.58 ± 1.90%
	2	140.163	151.9662	144.5892	147.54			
	3	149.0154	151.9662	147.54	146.0646			
Popcorn	1	100.41	95.38	103.412	98.392	100.14 ± 3.015 mL	130 mL	−22.97 ± 2.31%
	2	95.38	103.412	102.408	104.416			
	3	98.392	97.388	102.408	100.4			
Chocolate Brownie	1	135.44	131.81	131.81	134.5	133.90 ± 3.35 mL	135 mL	−0.81 ± 2.48%
	2	130.465	138.535	131.81	138.535			
	3	133.155	129.12	139.88	131.81			
Banana	1	96.192	97.194	98.196	99.198	100.16 ± 2.65 mL	120 mL	−16.57 ± 2.20%
	2	103.206	99.198	101.202	105.21			
	3	97.194	103.206	101.202	100.2			
Mean Absolute Error:								16.62%

**Table 4 sensors-22-03300-t004:** Errors in volume estimation using FOODCAM.

Food Item	Random Position	Predicted Volume (mL)				Mean Predicted Volume (Mean ± Std. Dev.)	Ground Truth	Error in Volume Estimation (Mean ± Std. Dev.)
		Trial 1	Trial 2	Trial 3	Trial 4			
Chickpeas	1	103.47	96.27	89.92	88.84	93.77 ± 5.33 mL	100 mL	−6.23 ± 5.33%
	2	101.99	100.24	86.74	91.87			
	3	92.79	94.23	89.04	89.82			
French Fries	1	166.22	164.76	163.15	165.40	169.67 ± 8.04 mL	180 mL	−5.73± 4.46%
	2	165.65	164.85	190.07	166.98			
	3	166.07	180.80	164.98	177.18			
Popcorn	1	134.61	123.65	127.80	137.74	137.53 ± 7.95 mL	130 mL	5.79 ± 6.12%
	2	137.98	149.75	143.44	150.04			
	3	142.30	136.61	127.67	138.83			
Chocolate Brownie	1	141.29	137.32	142.12	136.03	143.88 ± 5.46 mL	135 mL	6.58 ± 4.04%
	2	148.91	152.29	143.86	149.20			
Banana	1	108.86	118.38	130.28	117.82	115.58 ± 9.74 mL	120 mL	−3.68 ± 8.11%
	2	126.28	118.29	125.67	124.72			
	3	102.03	103.44	101.94	109.28			
Mean Absolute Error:								5.60%

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
