# Peer review of "FOODCAM: A Novel Structured Light-Stereo Imaging System for Food Portion Size Estimation"

_sensors, 2022, doi:10.3390/s22093300_

Round 1

Reviewer 1 Report

In this manuscript, Raju et al. designed a novel passive, standalone, multispectral, motion-activated, structured light-supplemented, stereo camera for food intake monitoring (FOODCAM) and an associated methodology for Food Portion Size Estimation. Furthermore, the method combined two state-of-the-art volume estimation methods and did not require fiducial markers. In general, this work is interesting; the test results with high accuracy suggesting FOODCAM can potentially be used as as a reliable method for FPSE. In addition, the manuscript is well presented with proper discussion. I would recommend its publication after the following minor issues are addressed:

  1. In Table 4, the Mean Absolute Error was calculated to be 5.60%, however this value was declared as 5.40% in Conclusion Section. The authors should check this data and update the average accuracy (94.6%) in Abstract Section.

Author Response

Point 1: In Table 4, the Mean Absolute Error was calculated to be 5.60%, however, this value was declared as 5.40% in Conclusion Section. The authors should check this data and update the average accuracy (94.6%) in Abstract Section.

Response 1: We greatly appreciate the reviewer for pointing out the error in the reported accuracy. We have fixed this oversight and corrected the abstract and the conclusion section. The Reported error of 5.60 % in the table is the correct value and we have used the same throughout the rest of the paper. 

Reviewer 2 Report

The authors present a new device for 3D reconstructions and volume estimation of objects. It is based on an RGB+IR stereo pair and an IR projector to produce a structured light pattern, which is employed to enhance 3d reconstruction. The performance of the device is evaluated by measuring the volume of plastic food models. The measurements obtained are compared against a commercially available device for reference. According to the authors evaluation, the proposed device provides measurements with an average error around 5.6%.

The paper is easy to read, however, I would recommend some major revision to improve the quality of this work.  Before I could recommend publication, I would like the authors to answer the following comments:

  1. My main concern with this work is novelty. There are already commercial products with similar characteristics. Such is the case of Intel RealSense d400 series which are low cost and high precision devices, which even implement autocalibration procedures. These devices also employ a stereo pair backed up with structured light. I would like the authors to comment on the advantages of your proposal. Furthermore, can you comment on the comparison of your device against other ToF and Stereo-pairs proposals in the state of the art?
  2. The experiments were performed using five food models. According to Table 3 and 4, each model was imaged 4 times in 3 different positions. Therefore, the reported errors are based in 12 sample measurements only. I’m worried about repetition of the results. If possible, increase the sample size of your experiments. Plus, why is there no std. dev reported in the mean error in volume estimation for Pico Flexx Cam in Table 3?
  3. The paper needs to explicitly state its contribution. The authors dedicate section 2 to material and methods, it looks like they employ well know techniques and procedures. However, it is not clear what is the novel contribution from this project to the state of the art.
  4. In the discussion section, the authors mention the following advantage “the proposed approach doesn’t require any fiducial markers reducing user burden”. I don’t think this is quite correct, since according to the results section, the user must manually segment the ROIs using the RGB data. I believe this implies a higher burden on the user.

Round 2

Reviewer 2 Report

The authors present a new device for 3D reconstructions and volume estimation of objects. It is based on an RGB+IR stereo pair and an IR projector to produce a structured light pattern, which is employed to enhance 3d reconstruction. The performance of the device is evaluated by measuring the volume of plastic food models. The measurements obtained are compared against a commercially available device for reference. According to the authors evaluation, the proposed device provides measurements with an average accuracy around 94%.

The paper is easy to read, and the authors have improved the quality of the paper.  I have no further comments but to recommend publication.